# Development of Novel Polysaccharide Membranes for Guided Bone Regeneration: In Vitro and In Vivo Evaluations

**DOI:** 10.3390/bioengineering10111257

**Published:** 2023-10-28

**Authors:** Naïma Ahmed Omar, Jéssica Roque, Paul Galvez, Robin Siadous, Olivier Chassande, Sylvain Catros, Joëlle Amédée, Samantha Roques, Marlène Durand, Céline Bergeaut, Laurent Bidault, Paola Aprile, Didier Letourneur, Jean-Christophe Fricain, Mathilde Fenelon

**Affiliations:** 1Laboratory for Tissue Bioengineering, University of Bordeaux, INSERM 1026, F-33076 Bordeaux, France; naima.ahmed-omar@u-bordeaux.fr (N.A.O.); olivier.chassande@inserm.fr (O.C.); mathilde.fenelon@u-bordeaux.fr (M.F.); 2Department of Oral Surgery, University Hospital of Bordeaux, F-33076 Bordeaux, France; 3Centre d’Investigation Clinique de Bordeaux (CIC 1401), University Hospital of Bordeaux, INSERM, F-33000 Bordeaux, Francemarlene.durand@chu-bordeaux.fr (M.D.); 4Siltiss, SA, Zac de la Nau, 19240 Saint-Viance, France; c.bergeaut@silab.fr (C.B.); l.bidault@silab.fr (L.B.); 5Laboratory for Vascular Translational Science (LVTS), X Bichat Hospital, University Paris Cité & University Sorbonne Paris Nord, INSERM 1148, F-75018 Paris, France

**Keywords:** guided bone regeneration, bioresorbable membrane, polysaccharide, pullulan, dextran, in vivo

## Abstract

Introduction: Guided bone regeneration (GBR) procedures require selecting suitable membranes for oral surgery. Pullulan and/or dextran-based polysaccharide materials have shown encouraging results in bone regeneration as bone substitutes but have not been used to produce barrier membranes. The present study aimed to develop and characterize pullulan/dextran-derived membranes for GBR. Materials and methods: Two pullulan/dextran-based membranes, containing or not hydroxyapatite (HA) particles, were developed. In vitro, cytotoxicity evaluation was performed using human bone marrow mesenchymal stem cells (hBMSCs). Biocompatibility was assessed on rats in a subcutaneous model for up to 16 weeks. In vivo, rat femoral defects were created on 36 rats to compare the two pullulan/dextran-based membranes with a commercial collagen membrane (Bio-Gide^®^). Bone repair was assessed radiologically and histologically. Results: Both polysaccharide membranes demonstrated cytocompatibility and biocompatibility. Micro-computed tomography (micro-CT) analyses at two weeks revealed that the HA-containing membrane promoted a significant increase in bone formation compared to Bio-Gide^®^. At one month, similar effects were observed among the three membranes in terms of bone regeneration. Conclusion: The developed pullulan/dextran-based membranes evidenced biocompatibility without interfering with bone regeneration and maturation. The HA-containing membrane, which facilitated early bone regeneration and offered adequate mechanical support, showed promising potential for GBR procedures.

## 1. Introduction

Guided bone regeneration (GBR) procedure has been widely used for its successful and reliable outcomes to repair damaged bone in various clinical indications (i.e., oral or orthopedic bone defects caused by infection, malformation, traumatism, or tumor) [1,2]. This procedure relies on the use of a membrane that is applied between the surrounding soft tissue and the bone defect [3]. This barrier membrane prevents soft tissue invasion into the bone defect area, thereby allowing osteoprogenitor cells to preferentially fill the bone defect and proliferate to achieve bone regeneration [4].

GBR membranes may require several criteria for optimal bone regeneration such as (i) biocompatibility, (ii) space maintaining, (iii) appropriate resorption time, and (iv) easy handling and shaping [5,6]. Two main types of GBR membranes can be used in clinic: non-resorbable and resorbable membranes [4,7,8]. Despite the wide choice of existing barrier membranes available on the market, they all showed limitations. Non-resorbable synthetic membranes provide good mechanical strength to maintain bone growing space. However, they must be removed at a later stage, which entails risks of morbidity and pain for patients. In addition, membrane exposure is often reported to lead to bacterial contamination and subsequent infection [5,9]. To overcome these issues, resorbable membranes, from synthetic or natural components can be used. Unfortunately, their resorption rate and poor mechanical properties, leading to membrane collapsing, are often considered limiting factors in their use by surgeons [10].

One of the most commonly used resorbable membranes is made from collagen [11,12]. Collagen membranes are frequently of bovine or porcine origin. Nowadays, animal-derived membranes could be a risk for safety and a serious limitation for patients having moral commitments [13]. The need to develop new membranes for GBR applications is thus critical, and the use of natural animal-free polymers can even avoid ethical concerns [7,14,15]. 

Polysaccharides are one type of natural polymers composed of glucosidic units [16] and their application in tissue engineering is emerging [17,18,19]. In particular, pullulan and dextran are two polymers that showed promising results in the field of bone regeneration since they share similarities with bone extracellular matrix components. Their use in a mixed composition (75% Pullulan, 25% Dextran) as bone graft substitutes containing the mineral component hydroxyapatite (HA) has been extensively investigated [20,21,22,23,24]. These polysaccharides are used as injectable hydrogels (microbeads) that easily fit the bone defect. Pullulan/dextran-based microbeads have shown great osteogenic properties in vivo using a femoral condyle defect in rats, and in sinus floor augmentation procedures in sheep [20,21,22]. It has also been established that these materials, free from osteogenic cells, are able to retain growth factors, thus boosting osteogenesis [19]. 

To our knowledge, only one study by Miyahara et al. suggested the use of pullulan to provide a barrier membrane for GBR application [19,25]. Compared to a commercially available collagenic membrane, the pullulan-derived membrane showed higher bone formation after four weeks in a mice calvaria bone defect. Despite these promising results, no studies have investigated the production of a membrane derived from both pullulan and dextran biomaterials [19]. The aim of this study was to develop and characterize in vitro and in vivo two different formulations of pullulan/dextran-based membranes for GBR applications.

## 2. Materials and Methods

### 2.1. Materials 

Membranes were molded (15 cm length and width, 500 µm depth) as porous materials for guided bone regeneration according to an extensively published patented protocol for porous polysaccharide-based scaffold fabrication [20,21,22]. Briefly, 75% pullulan/25% dextran solutions mixed with sodium trimetaphosphate (2.5%) as a crosslinker were poured on a plate and freeze-dried to obtain films as previously described for porous discs, cylinders, or conical scaffolds. In this study, macro-sized hydroxyapatite particles were dispersed or not into the solutions of polysaccharides to obtain two new pullulan/dextran-based membranes (Mb) with or without hydroxyapatite particles (Mb and Mb + HA). 

To visualize the microstructure of both Mb and Mb + HA, scanning electron microscopy was performed. Each side of the freeze-dried membranes was first coated with a thin layer of gold before being observed using a scanning electron microscope (TM4000Plus, Hitachi, Tokyo, Japan) at an accelerating voltage of 15 kV. 

The mechanical properties of both membranes were investigated using nanoindentation assessment. First, membranes shaped into 6 mm circles and were hydrated 72 h in physiological serum (0.9% NaCl) prior to measurements. Local Young’s moduli were computed for each side of the membranes and repeated on three different samples (for each membrane). Local mechanical properties of membranes were measured with a Piuma Nanoindenter (Optics11, Amsterdam, The Netherlands), using an optical fiber connected to an interferometer and linked to a spherical probe (stiffness: 0.47 N·m^−1^; tip radius: 102 µm) to acquire force–displacement curves. To measure the Young’s modulus of the membranes, the probe was immersed for 6 μm into the sample at each point of measurement (25 points for each side and sample). The Young’s modulus for each point was computed according to the Hertz contact model for a spherical body indenting a flat surface, using the built-in Piuma software (version 2.5.0, Optics11, Amsterdam, The Netherlands), and the given mean of elastic modulus values was assessed.

### 2.2. In Vitro Cytotoxicity Assays

Human bone marrow mesenchymal stem cells (hBMSCs) were used for indirect cytotoxicity studies following the ISO standard 10993-5:2009 [26]. The hBMSCs were isolated from consenting patients who had undergone hip surgery and expanded according to well-established protocols [27]. Cells were used at passage 2 for this study and cultured in α-Minimum Essential Medium (α-MEM, GIBCO^®^) under a controlled atmosphere (5% CO_2_). Sample extracts (Mb and Mb + HA), positive and negative controls were prepared according to the ISO standard 10993-12:2021 [28]. Briefly, 0.1 g of the membrane (corresponding approximately to a square of 1 × 1 cm^2^) was incubated in 1 mL of α-MEM supplemented with 10% (*v*/*v*) fetal bovine serum for 72 h at 37 °C, 5% CO_2_. The negative control was the α-MEM and the positive control was prepared using α-MEM medium with 20% dimethyl sulfoxide (DMSO). The obtained supernatants were then placed in contact with the hBMSCs (3 × 10^4^ cells/cm²) for 24 h in 96-well plates to assess cell viability and metabolic activity (*n* = 6 per condition). 

Cell viability was performed using a Neutral Red (NR) test. Culture medium was removed from the plate and 50 µL of NR solution (diluted at 1.25% in IMDM medium) was added in each well and incubated for 3 h at 37 °C, 5% CO_2_. Supernatants were removed and wells were washed with HBSS (GIBCO^®^) to remove the excess of NR solution. Cells were lysed with 1% acetic acid in 50% ethanol. The resulting optical density (OD) values were read at 540 nm using a spectrophotometer (Varioskan Lux, Thermoscientific, Waltham, MA, USA). 

Metabolic activity was assessed using 3-[4,5-dimethyltriazol-2-y1]-2,5-diphenyl tetrazolium bromide (MTT) assay. As previously described for the NR test, the culture medium was removed and 100 µL of MTT solution (prepared from a stock solution of 5 mg/mL in 0.1 M PBS and diluted at 20% in IMDM medium without phenol red) was added in each well. After 2 h of incubation at 37 °C under a controlled atmosphere, supernatants were discarded and 100 µL of dimethyl sulfoxide (DMSO) was added to the wells to dissolve the formed crystals. To read the OD values, the plate was read at 570 nm using a spectrophotometer (Varioskan Lux, Thermoscientific, Waltham, MA, USA). 

For each assay, results of each condition were normalized to negative controls (cells cultured in α-MEM) as follows: OD = (OD_sample_/OD_control_) × 100. 

### 2.3. In Vivo Experiments

The two rat experimental procedures followed the principles of Laboratory Animal Care formulated by the National Society for Medical Research and approved by the Animal Care and Experiment Committee of the University of Bordeaux, Bordeaux, France. Experiments were carried out in an accredited animal facility following European recommendations for laboratory animal care (directive 86/609 CEE of 24/11/86). 

#### 2.3.1. Biocompatibility Assessment

-Surgical procedure:

A rat subcutaneous model was first used to assess the biocompatibility of the two different membrane formulations—Mb and Mb + HA. They were also compared to a commercially available and widely used collagen membrane (Bio-Gide^®^, Geistlich, Roissy-en-France, France) for GBR procedures. This study was approved by the French Ethics Committee under APAFIS n°2016030408537165v4. Fifteen Sprague Dawley male rats (11 weeks old) were used for subcutaneous implantation. Analgesia was provided by intraperitoneal injection of buprenorphine (0.05 mg/kg body weight, Buprécare^®^, Axience, Pantin, France) and rats were anesthetized by inhalation of isoflurane (induced with 4% and maintained at 2%). The back of the rats was shaved and disinfected with Betadine. Four horizontal incisions of 1 cm in length were made in the rat’s back and dorsal subcutaneous pockets were created via dissection. Within the same rat, four conditions were attributed to the subcutaneous pockets: (1) Mb, (2) Mb + HA, (3) Bio-Gide^®^, and (4) a sham-operated site. All membranes were shaped as 1 × 1 cm^2^ squares and hydrated in 0.9% NaCl for 5 min before implantation. Finally, the skin was closed with surgical clips and an antiseptic spray (Aluspray^®^, Vétoquinol, Lure, France) was applied to the scar. After surgery, food and water were supplied ad libitum. At 1, 4, and 16 weeks, rats were sacrificed (*n* = 5 per condition and per time) by carbon dioxide overdose. Subcutaneous implants were fixed in 4% paraformaldehyde (PFA) overnight at 4 °C.

-Histological analysis:

Samples were dehydrated with increasing concentrations of ethanol baths before paraffin embedding. Transversal cutting of rat skin was performed to obtain 7 µm slices. Two slides per condition were stained via HES staining and the images were acquired on a slide scanner (Nano-zoomer 2.0, Hamamatsu Photonics, Massy, France). Biocompatibility was assessed following the ISO standard 10993-6:2016 [29] by a blinded independent trained investigator. Inflammatory reaction was scored around the implants semi-quantitatively. This score was computed by investigating the cell type/response and the tissue response: cellular infiltration and inflammatory cell type (polymorphonuclear cells, lymphocytes, macrophages, plasma cells, and giant cells), vascularization, fatty infiltration, and extent of fibrosis. Membranes were compared to the sham-operated control samples and were considered as non-irritant for a score from 0.0 up to 2.9, slightly irritant (3.0 up to 8.9), moderately irritant (9.0 up to 15.0) or severely irritant (>15) to the tissue [30]. Finally, membrane resorption was assessed based on membrane thickness measurements over time using histological sections on the NDP-View software version 2.0 (Hamamatsu Photonics, Massy, France).

#### 2.3.2. Femoral Defect Implantation 

-Surgical procedure:

The potential of these polysaccharide membranes (Mb and Mb + HA) to act as a barrier for GBR procedures was assessed using a non-critical femoral bone defect model [31]. Additionally, they were compared to Bio-Gide^®^ (Geistlich, Roissy-en-France, France). The study was approved by the French Ethics Committee under APAFIS n° 32504-2021072111152646v4. Thirty-six Sprague Dawley male rats (11 weeks old) were used for the femoral defect implantation (two defects per rat). Briefly, both legs of each rat were shaved and disinfected with Betadine. Analgesia was given via intraperitoneal injection of buprenorphine (0.05 mg/kg body weight, Buprécare^®^), and rats were anesthetized via inhalation of isoflurane (induced with 4% and maintained at 2%). A longitudinal skin incision was made laterally across both legs, and muscles were dissected to expose the femoral diaphysis. The periosteum was removed manually. A 2.3 mm bone defect was then drilled under irrigation. The defect was either left empty or covered by a membrane: (1) Mb, (2) Mb + HA, and (3) Bio-Gide^®^ (Geistlich France). Membranes were shaped as 2 × 1 cm^2^ rectangles to cover the whole femur and recut if necessary. They were maintained over the defect using a resorbable suture thread (4-0 Vicryl^®^, Ethicon, division of Johnson & Johnson, Brussels, Belgium). A suture thread was also applied for the empty defect for consistency. The muscles were subsequently repositioned and sutured with absorbable sutures (4-0 Vicryl^®^), and the skin was closed with surgical clips. An antiseptic spray (Aluspray^®^) was applied to the scar. After surgery, food and water were supplied ad libitum. Animals were sacrificed at 1, 2, and 4 weeks post-operation (*n* = 6 femurs per condition and per time) and fixed overnight in PFA 4% at 4 °C. 

-Radiographic analysis:

Micro-computed tomography (Quantum FX Caliper, Life Sciences, Perkin Elmer, United States) was performed ex vivo on femurs. The X-ray source was set at 90 kV and a current of 160 μA used to obtain a 10 μm resolution (field of view: 5 mm). After scanning, visualization of the cross-sectional slices was performed using the VGSTUDIO MAX^®^ software (version 2022.3, Volume Graphics, Heidelberg, Germany). To assess bone volume fraction (BV/TV, ratio between bone volume and total volume), 3D images were oriented to face the defect (coronal view). Then, a 2.3 mm-diameter, 550 μm-depth cylindrical volume of interest, corresponding to the initial surgical defect, was created. Each scan was reconstructed using the same calibration system to distinguish bone and air. 

-Histological analysis:

Each sample was decalcified with EDTA-based Microdec^®^ (Diapath, MM Brignais, Brignais, France) decalcifiant for three weeks under gentle agitation. Samples were then dehydrated with increasing concentrations of ethanol baths and processed for embedding in paraffin. Sagittal cuts of rat femoral bones were made to obtain 7 μm-thick serial sections in the middle of the defect. First, slides were stained with Masson–Goldner’s trichrome staining to perform histomorphometric analysis of newly formed bone. Then, slides were stained using picrosirius red staining to analyze the collagen fibers’ orientation under polarized light microscopy (Widefield microscope DM 5000, Leica, Nanterre, France). Orientation was assessed inside the defect and computed using the OrientationJ plug-in (ImageJ) to measure the main direction of the fibers [32]. This criterion was evaluated by computing the coherency that shows if the local image features are orientated or not (expressed here in percentage—0% indicates the image is isotropic and 100% indicates that the image has one dominant orientation). 

### 2.4. Statistical Analysis

Results were expressed as mean ± SD and “*n*” indicated the number of membranes tested. The GraphPad Prism Software 8.2.1. (La Jolla, CA, USA) was used to perform statistical analysis. A normality was first performed using a D’Agostino and Pearson omnibus normality test. Statistical significance between several groups was assessed via one-way analysis of variance (ANOVA) followed by Bonferroni post-test for data assuming a Gaussian distribution. Differences for independent samples were evaluated with the non-parametric Kruskal–Wallis test and Dunn’s multiple comparison test. The non-parametric Mann–Whitney test (two-tailed) was used to compare two groups. Differences were considered significant and indicated with a star when *p* < 0.05 (*), *p* < 0.01 (**), *p* < 0.001 (***), and *p* < 0.0001 (****).

## 3. Results

### 3.1. Materials

The morphology of the Mb and Mb + HA membranes as well as the collagenic membrane (Bio-Gide^®^) was observed using scanning electron microscopy (Figure 1). Bio-Gide^®^ demonstrated a bilayered structure with a smooth side composed of compact collagen fibers and a rough side with a fibrous open-pored pattern. Concerning Mb and Mb + HA, these porous pullulan/dextran 3D scaffolds have been extensively described by our team [33,34,35,36,37,38]. Both freeze-dried membranes have small pores (10–20 µm) on the smooth side. Mb is highly porous on the other side (i.e., rough side), as previously reported [34], with a mean pore size of 200–300 µm (Figure 1, middle row). The addition of HA influenced the crosslinking and pore formation resulting in smaller pores (30–80 µm) on the rough side for Mb + HA (Figure 1, bottom row) [23]. The pores are elongated with a mean Feret shape around 2 [35]. This allows the penetration of individual osteoblasts or larger spheroids [36,37]. Experimental and numerical simulations of the oxygen transport evidenced that the oxygen diffusion coefficient (D = 1.6 ± 0.5 × 10^−9^ m^2^ s^−1^) in the polysaccharide scaffolds was favorable for cell viability [35]. Additionally, the degradation kinetics in vivo can be tuned to parallel the tissue regeneration [38,39]. Moreover, the mechanical properties of these membranes are in the range of connective tissues of mammals [38,39]. Surface mechanical properties of the membranes were evaluated here by nanoindentation. Both membranes showed comparable Young’s moduli regardless of the side of the membrane (values comprised between 4.73 and 8.95 kPa). These results are consistent with elastic moduli computed for the Bio-Gide^®^ membrane reinforcing the potential used of these pullulan/dextran membranes for GBR application.

### 3.2. In Vitro Cytocompatibility

Cell viability was carried out via MTT and Neutral Red assays (Figure 2). No cytotoxic effect was observed for both formulations, Mb and Mb + HA, since cell viability was up to 70% according to the ISO standard 10993-5 [26] and compared to the negative control (i.e., standard cell culture medium) (Figure 2a). Similar results were obtained for the evaluation of metabolic activity (Figure 2b). Mb and Mb + HA were considered non-cytotoxic, which is a necessary validation criterion before implanting them in vivo.

### 3.3. In Vivo Experiments

#### 3.3.1. Biocompatibility 

Four conditions (Sham, Bio-Gide^®^, Mb, and Mb + HA groups) in a rat subcutaneous model at different time points (1, 4 and 16 weeks) were evaluated (Figure 3a). A total of 97 samples were analyzed to evaluate the inflammatory reaction score following the ISO standard 10993-6 [29] using HES staining (Figure 3a). A blinded independent trained investigator evaluated cell type and tissue responses. After 1 week of implantation, a slight inflammatory reaction was observed for the three different membranes compared to the sham-operated control (Figure 3a). A fibrotic capsule appeared around the implants for Mb and Mb + HA at 1 week. It became thinner and compact after 4 weeks suggesting that it stabilized. For the Bio-Gide^®^ membrane, the slight inflammatory reaction remained at 4 weeks, whereas Mb and Mb + HA were considered as non-inflammatory at 4 weeks. A progressive cellular infiltration inside the Bio-Gide^®^ membrane was also observed, whereas no cell infiltration was noticed for Mb or Mb + HA. All membranes were non-irritant 16 weeks post-operation and were still visible in the subcutaneous tissue. Based on these findings, the inflammatory reaction score (Figure 3b) confirmed that Mb and Mb + HA were considered biocompatible biomaterials. 

Membrane thicknesses were measured to estimate their resorption over time (Figure 3c). Mb resorbed quickly up to 31.8% between 1 and 4 weeks but remained stable until 16 weeks (compared to 1 week). Mb + HA showed the slowest rate of resorption with a resorption rate of 10.8% between 1 and 4 weeks and up to 13.1% at 16 weeks. We also observed that the Bio-Gide^®^ membrane resorbed faster and to a greater extent over time (25.8% at 4 weeks, and 40% at 16 weeks).

#### 3.3.2. Bone Regeneration of Femoral Defects

Then, the osteogenic properties of membranes and their ability to act as a barrier were investigated in a non-critical size femoral defect among 36 rats. Micro-CT was performed after 1, 2, and 4 weeks of implantation for the four groups (Empty, Bio-Gide^®^, Mb, and Mb + HA) (Figure 4a). Bone regeneration (BV/TV) was quantified at 1, 2, and 4 weeks after the surgery using micro-CT analysis (Figure 4b). At 1 week after the surgery, no differences were observed between conditions (1 week BV/TV [%]: Empty = 11.62 ± 6.76, Bio-Gide^®^ = 6.87 ± 4.61, Mb = 12.79 ± 6.91, and Mb + HA = 12.79 ± 11.50). 

Both polysaccharide membranes revealed a comparable BV/TV as well as the empty defect 2 weeks post-operation. However, bone regeneration was significantly enhanced when the defect was covered with Mb + HA compared to Bio-Gide^®^ (2 weeks BV/TV [%]: Empty = 45.83 ± 14.65, Bio-Gide^®^ = 28.22 ± 12.78, Mb = 44.88 ± 16.48, and Mb + HA = 50.88 ± 12.69, *p* < 0.05). The late time point (4 weeks) did not enable the differentiation of the four conditions (4 weeks BV/TV [%]: Empty = 61.55 ± 11.73, Bio-Gide^®^ = 59.65 ± 15.06, Mb = 58.19 ± 8.40, and Mb + HA = 49.65 ± 10.74).

Histomorphometric analysis of Masson–Goldner’s trichrome staining was consistent with the micro-CT results and supported new bone formation inside the defect previously observed (Figure 5). Macroscopically, histological sections revealed that Mb and Bio-Gide^®^ tended to collapse inside the defect (Figure 5a), thereby reducing the bone healing space at the earliest stages of bone regeneration. This phenomenon was not observed with Mb + HA. Similar amounts of newly formed bone were observed after 1 week for the two polysaccharide membranes and the commercial membrane compared to the defect left empty (1 week newly formed bone [%]: Empty = 14.50 ± 7.85, Bio-Gide^®^ = 15.34 ± 5.69, Mb = 14.63 ± 3.94, and Mb + HA = 12.02 ± 3.41). The two polysaccharide membranes revealed a comparable amount of bone, as well as the empty defect two weeks after the surgery, compared to Bio-Gide^®^ membrane (2 weeks newly formed bone [%]: Empty = 41.09 ± 10.59, Bio-Gide^®^ = 28.36 ± 8.63, Mb = 46.62 ± 16.66, and Mb + HA = 42.88 ± 11.15). For the late time point, newly synthesized bone appeared well-organized with lamellar structure (i.e., parallel fibers) for the polysaccharide-based membrane conditions. To highlight this finding, bone quality at 4 weeks was assessed using picrosirius red staining to evidence fiber orientation and ultimately bone maturation (Figure 6). Fiber orientation was analyzed by reporting its coherency. Host bone was used as a reference for well-organized lamellar structure with coherency around 35% showing organized collagen fibers. The quality of newly formed bone was thus compared to the host bone. 

Covering the defect with Mb + HA resulted in the formation of bone with significantly better fiber orientation and organization compared to Mb alone (*p* < 0.05) with a coherency up to 14.68 ± 3.52% and 7.83 ± 2.62%, respectively. Comparable fibers orientation was seen for Bio-Gide^®^ and Empty groups (Coherency [%]: Empty = 10.13 ± 3.67 and Bio-Gide^®^ = 10.03 ± 3.54).

## 4. Discussion

Nowadays, resorbable membranes are widely used to overcome the major disadvantages of non-resorbable membranes for GBR. These membranes are mainly derived from animals (e.g., collagen) making them unfit to use in patients with moral constraints [13]. Additionally, their poor mechanical stability and uncontrolled degradation rates are a limitation [40]. The present study aimed to characterize the biocompatibility and the osteogenic potential of two natural membranes for GBR procedures made from pullulan and dextran and containing or not HA. Dextran and pullulan are two natural exopolysaccharides that have already been used for medical purposes [15,19]. The association of both polysaccharides to design bone substitutes showed promising results in the field of bone regeneration [23]. This is the first study to design pullulan/dextran-derived membranes and assess their in vitro and in vivo performances for GBR application.

Four requirements to assess successful GBR were described by Wang et al. and are known under the “PASS” principles: (i) primary wound closure, (ii) angiogenesis for adequate blood supply, (iii) space maintenance of newly formed bone, and (iv) wound stability to allow blood clot formation [7,41].

When considering a material for human implantation, biocompatibility is one of the most important factors to be ensured. Polysaccharides are one type of natural polymers that show high biocompatibility and similar structure compared to extracellular components [42]. The performance of the pullulan/dextran membranes was assessed in vitro for their cytotoxicity and in vivo on ectopic and orthotopic rat models. First, the cytocompatibility of both membranes was evaluated in vitro on hBMSCs to closely reproduce clinical conditions. Both pullulan/dextran-derived membranes were cytocompatible. This is consistent with results obtained from previous studies where hBMSCs were seeded on pullulan/dextran-derived scaffolds with successful proliferation rate for up to 15 days [21]. 

To further investigate their biocompatibility, a rat subcutaneous implantation model was then conducted. The in vivo subcutaneous implantation model is known to provide critical information on the host tissue reaction of barrier membranes, as well as their tissue integration patterns, by assessing cellular infiltration, vascularization, and degradation [43]. Blinded histological analysis revealed that both polysaccharide membranes showed slight acute inflammatory reaction after one week, and were considered as non-irritant four weeks after implantation, thereby indicating their biocompatibility. This is corroborated by a previous study that investigated the biocompatibility of hydrogels with varying concentrations of dextran and pullulan [44]. In that study, a noticeable inflammation was observed 30 days after implantation when pullulan was implanted alone, whereas the implantation of dextran-based hydrogel alone led to a higher fibrous capsule formation that could induce post-implantation pain for patients [45]. Interestingly, the study highlighted the interest of combining pullulan and dextran to improve the biocompatibility by decreasing the foreign body reaction. Thus, the ratio of 75/25 for pullulan and dextran, respectively, provided the best candidate for further investigations.

A slowest resorption rate was observed with the pullulan/dextran-derived membranes compared to the commercial collagen membrane. This finding was also supported by previous studies reporting the tendency of collagen membranes to quickly resorb [43,46,47,48]. A possible explanation is the progressive cellular infiltration observed inside the Bio-Gide^®^ membrane, whereas no cell infiltration was observed inside Mb and Mb + HA. Membranes were designed to act as a physical barrier to prevent surrounding tissue invasion inside the defect. 

Moreover, both pullulan/dextran-derived membranes remained up to 16 weeks after implantation, thereby making them attractive candidates for GBR procedure since clinical requirements specified their need for space maintenance between 16 and 24 weeks [1,10]. It should be emphasized that the pullulan/dextran membrane containing HA showed the lowest resorption rate. As previously reported by Piattelli et al., it could be explained by the addition of the mineral component (HA), thereby providing a slowest degradation to the polysaccharide membrane containing HA [49]. Despite its proven stability until 16 weeks, the membrane degradability must be assessed using longer timepoints (e.g., 6 to 12 months).

The potential osteogenic properties of the membranes were then investigated using a rat mono-cortical femoral bone defect. Both quantitative and qualitative bone formation analyses were performed. The results showed a significant enhancement of early bone regeneration using a pullulan/dextran-based membrane containing HA compared to the commercial membrane Bio-Gide^®^ due to better mechanical support. Indeed, the progressive mineralization of Mb + HA may increase its ability to support bone ingrowth and avoid the collapsing of the surrounding tissues. Huang et al. highlighted in a study the improvement of the mechanical properties of a chitosan membrane incorporated with chitosan microspheres loaded with in situ HA [50]. An investigation of the membrane mechanical properties ex vivo could provide valuable information to demonstrate the space maintenance capacity of Mb + HA as one of the main biological features desired for GBR procedure [41]. We also observed during the surgical procedure that Mb + HA was easier to handle and manipulate. We thus hypothesized that Mb + HA might offer better space-maintaining ability, thereby favoring an increase in bone formation. 

This hypothesis was corroborated by histological sections that showed early collapsing inside the defect when using the polysaccharide membrane without HA and the Bio-Gide^®^ membrane, whereas collapsing was not observed with Mb + HA. Therefore, the addition of mineral particles enhances space-maintaining properties, which is critical to guarantee the blood clot’s stability and then to support bone formation accordingly. 

The organization and spatial distribution of collagen fibers contained in the newly formed bone was also assessed, since they play a critical role in guiding its mechanical and biological properties [51]. Picrosirius red staining was thus performed to assess fiber orientation. This method, relying on the birefringence of collagen fibers, is used in combination with polarized light microscopy to specifically highlight collagen networks [52,53]. Our results showed that covering the defect with Mb + HA improved bone quality regarding collagen fibers orientation and organization compared to the membrane without HA. This finding suggested the potential use of this membrane for GBR procedures to maintain bone substitutes inside the defect area (e.g., femoral segmental defect). 

Our analysis of late bone regeneration did not evidence differences between conditions. This could be explained by the defect model in this study, a non-critical size defect, thereby showing bone healing in all the conditions after one month. In addition, longer timepoints should be considered (e.g., 8 weeks) to investigate bone healing quality and enable different healing processes between conditions. However, this mono-cortical femoral bone defect model was chosen mainly for two reasons. First, non-critical size femoral bone defects are widely performed in rodents because it is known to be an easily replicable and reliable model to perform, which is also easier to quantify [31,54,55]. Furthermore, the healing process has similarities to that of the jaw bones, which occurs through intramembranous ossification [54]. The present study mainly aimed to assess two pullulan/dextran-based membranes with or without HA to select the one with the best potential for GBR. From all the observed data obtained here, Mb + HA seemed to be the best candidate as it offered a better space-making property to enable bone growth and was easier to handle for the surgeon. The selected Mb + HA will be further assessed in a larger maxillary pre-clinical model in association with bone graft substitutes [56,57]. To further support the relevance of our material, it should be tested in a critical size defect in rats (e.g., mandibular defect). Additionally, to promote its translation into clinics, a larger animal model should be considered (e.g., sinus floor augmentation in sheep) [21,22]. This model will enable the assessment of the potential of the membrane in a more relevant model that (i) better mimics the anatomical and compositional characteristics of human bones and that (ii) it is easier to perform in a clinical setting due to its manageability.

Further investigations on the polysaccharide membranes morphology need to be studied to better understand their GBR properties. Indeed, the Bio-Gide^®^ membrane is composed of two distinct sides: (i) a rough-like side with an open-pore layer to face the defect and support osteogenic cell proliferation and vascularization, and (ii) a smooth-like side with a compact collagen layer to limit soft tissue invasion [58]. For the polysaccharide membranes, the influence of porosity will be a key component to ensure occlusive effect at the defect site since the membrane should limit cell invasion without compromising the oxygen and nutrient exchanges [1,33,59]. 

## 5. Conclusions

This study is the first to develop and evaluate pullulan/dextran-based membranes, associated or not with HA, for GBR procedures. Both membranes fulfilled several fundamental criteria for effective barrier membrane. They were not only non-cytotoxic in vitro but also biocompatible in vivo. The incorporation of HA into the membrane formulation improved mechanical handling and space-maintaining ability to guide bone formation. Mb + HA also enhanced early bone regeneration and favored the formation of a well-organized and mature bone. This membrane achieved the most promising results for GBR procedures. Future investigations will question its translation into clinical practice. To do so, it will be implanted in a larger maxillofacial pre-clinical model in association with bone graft substitutes to further assess its efficacy and potential for enhancing GBR procedures.

## Figures and Tables

**Figure 1 bioengineering-10-01257-f001:**
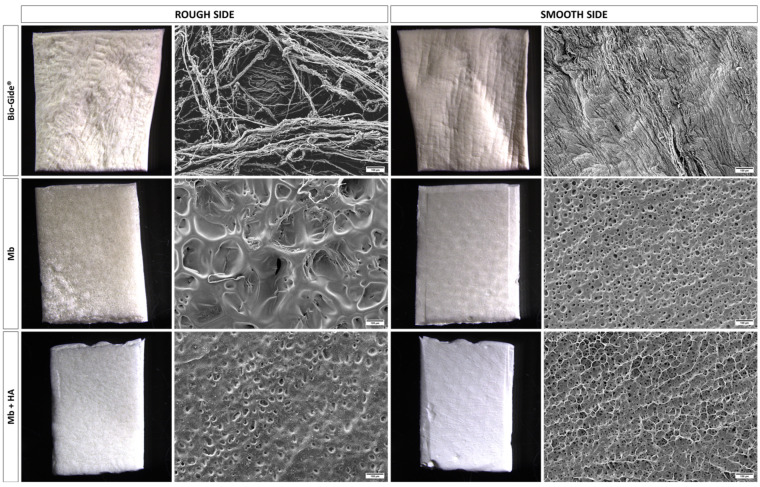
Membranes morphology. Macroscopic pictures of membranes showing the two sides of the materials and their corresponding SEM images (scale bar: 100 µm).

**Figure 2 bioengineering-10-01257-f002:**
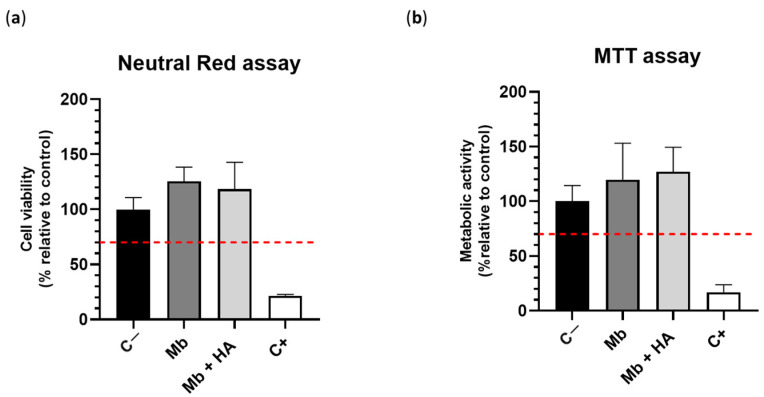
Cytotoxic effect of the membranes in vitro on hBMSCs according to the ISO standard 10993-5 (*n* = 6/condition); (**a**) MTT assay; (**b**) Neutral red assay; C− negative control, C+ positive control. The red doted horizontal lines corresponded to the cytocompatibility threshold.

**Figure 3 bioengineering-10-01257-f003:**
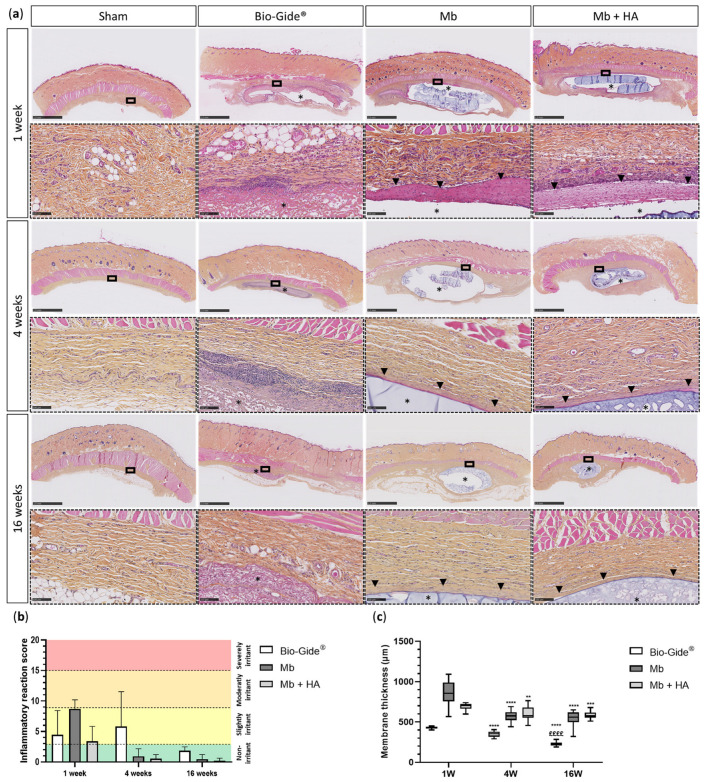
Assessment of biocompatibility using a rat subcutaneous model. (**a**) Histological section of subcutaneous implantation of the membranes via HES staining (low magnification, scale bar: 2.50 mm; high magnification, scale bar: 100 µm); black asterisks represent the remaining membrane (*n* = 6−10 per condition and timepoint), black arrowheads represent the thin fibrotic layer; (**b**) inflammatory reaction scores of the corresponding membranes; (**c**) thickness measurement over time, one-way ANOVA, ** *p* < 0.01, *** *p* < 0.001, **** *p* < 0.0001 compared to 1 week; ££££ *p* < 0.0001 compared to 4 weeks.

**Figure 4 bioengineering-10-01257-f004:**
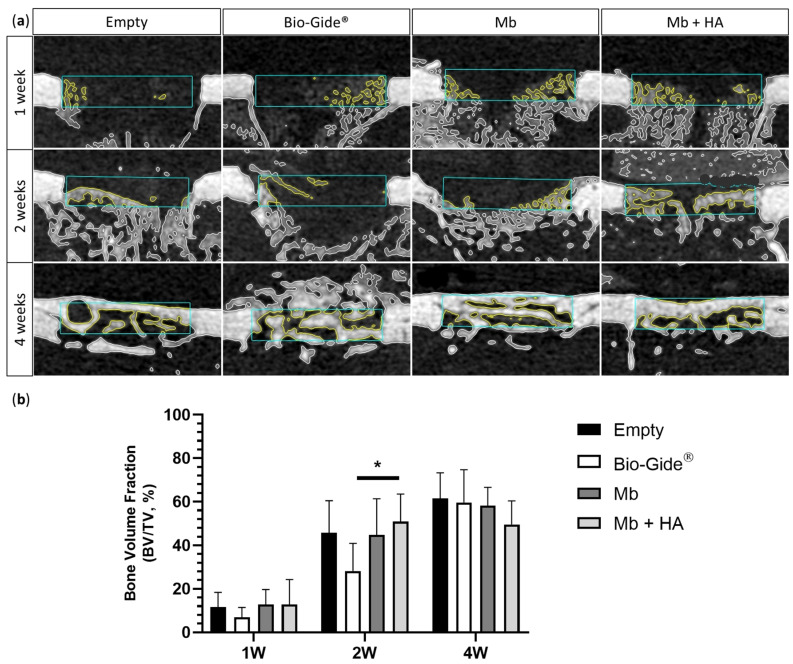
Rat femoral bone defect performed in the middle of the diaphysis. (**a**) 2D micro-computed tomography scans showing approximately in the middle of the defect, the blue rectangles represent the analyzed volume of interest; (**b**) bone volume fraction measurement using VGSTUDIO MAX software version 2022.3 (*n* = 6 per condition and time, except for Mb + HA at 2 weeks and Bio-Gide^®^ at 4 weeks, *n* = 5), Mann–Whitney test * *p* < 0.05.

**Figure 5 bioengineering-10-01257-f005:**
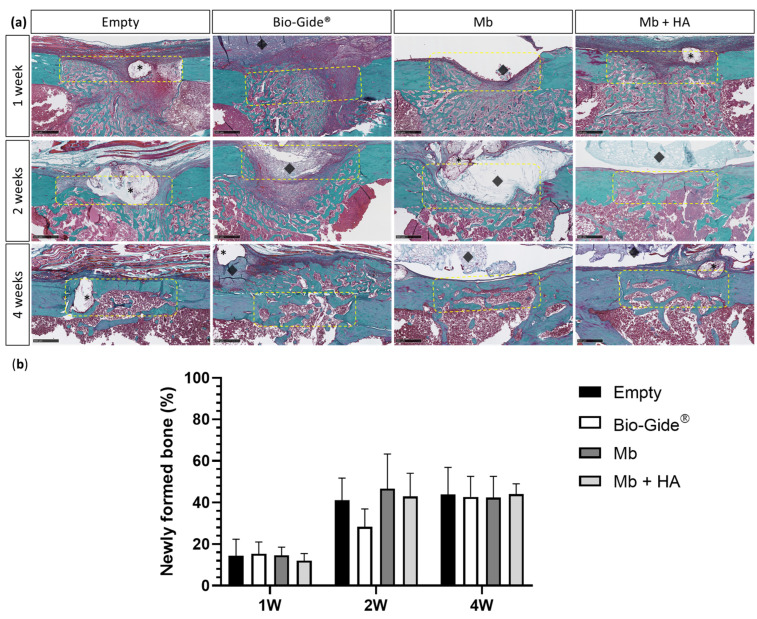
Histological analysis of the rat femoral bone defects for all groups (Empty, Bio-Gide^®^, Mb and Mb + HA) at 1 week, 2 weeks, and 4 weeks of explantation. (**a**) Masson–Goldner’s staining of the slides for the defects left empty or covered by a membrane (scale bar: 500 µm). Black asterisks represented the suture thread, the yellow dotted rectangles the bone defect, and black diamonds the remaining membrane; (**b**) histomorphometric analysis of the corresponding histological sections for newly formed bone.

**Figure 6 bioengineering-10-01257-f006:**
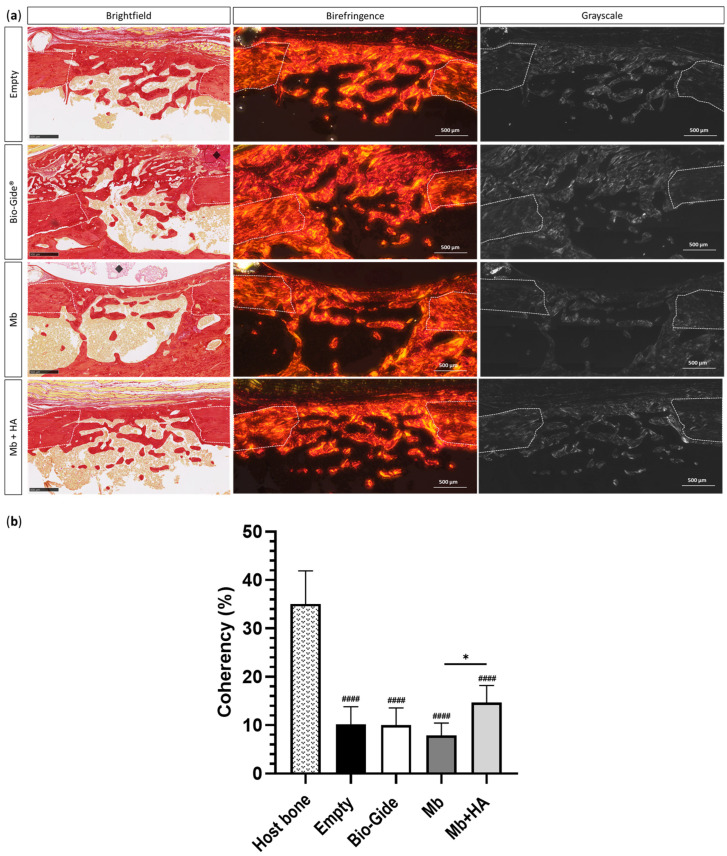
Picrosirius red staining of rat bone femoral defects for all groups (Empty, Bio-Gide^®^, Mb and Mb + HA) at 4 weeks. (**a**) Histological sections under brightfield view and polarized light view (birefringence and grayscale modes) of the slides for the defects left empty or covered by a membrane (scale bar: 500 µm). White dotted lines represented the edges of the defect, and the black diamonds represented the remaining membrane; (**b**) histomorphometric analysis of the corresponding histological sections. Collagen fibers orientation using OrientationJ plug-in (ImageJ), * *p* < 0.05, #### *p* < 0.0001, compared to host bone.

## Data Availability

The data that support the findings of this study are available from the corresponding author upon reasonable request.

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
