# Peer review of "Development of Novel Polysaccharide Membranes for Guided Bone Regeneration: In Vitro and In Vivo Evaluations"

_bioengineering, 2023, doi:10.3390/bioengineering10111257_

Round 1
Reviewer 1 Report
Development of novel polysaccharide membranes for Guided Bone
Regeneration: In vitro and in vivo evaluations
This research was addressed “to develop and characterize in vitro and in
vivo two different formulations of pullulan/dextran-based membranes for
GBR applications”.
This topic is not totally original because the authors said that “the
animal-derived membranes could be a risk for safety, and a serious
limitation for patients having some moral commitments and for ethics
reason”; for this reason consider is necessary to develop new membranes
for GBR. However, nowadays there are several types of membranes.
It would be necessary to explain why this type of polysaccharide would
be eligible over the existing types.
In general, the preparation of the membrane is based on previous studies
and some modifications made by the researchers are added.
The experimentation is sound but more studies are necessary to prove
that the thin fibrotic capsule has appeared around the implants treated
with Mb and Mb + HA, not represent a risk in the future
The conclusions are consistent but the study is an early stage
Author Response
Dear Reviewer,
Please consider the responses to the reviewers of our manuscript entitled “Development of novel polysaccharide membranes for Guided Bone Regeneration: in vitro and in vivo evaluations”.
We thank the reviewers for their thorough revision and we consider that the reviewer’s comments have helped to improve substantially the manuscript. We have modified the manuscript accordingly, as explained in detail in the responses to the reviewer and as highlighted in yellow in the main manuscript file (please find the attached document).
We hope that the changes we have made to the manuscript and our answers are in accordance with the reviewer requirements and, as a consequence, that this manuscript could be accepted for publication in “Bioengineering”.
Should you require any additional information and/or clarification, we remain available to provide it to you.
With our best regards,
Naïma AHMED OMAR
----------------------------------------------------------------------------------------
Comments
This topic is not totally original because the authors said that “the animal-derived membranes could be a risk for safety, and a serious limitation for patients having some moral commitments and for ethics reason”; for this reason consider is necessary to develop new membranes for GBR. However, nowadays there are several types of membranes. It would be necessary to explain why this type of polysaccharide would be eligible over the existing types.
Thank you for this valuable comment. There are indeed different types of resorbable membranes that are being used in practice and derived from either synthetic (e.g., polylactic acid, polylactic-co-glycolic acid) or natural components (e.g., collagen from bovine or porcine origin). Extensive research on the development of new GBR membranes is emerging, however, they are still under preclinical validation.
Despite the broad use of collagen membranes in clinics, the development of new biomaterials is needed to overcome the limitations encountered (e.g., mechanical property, uncontrolled degradation rate, moral concerns, and ethical issues).
Here, we proposed the use of two polysaccharides (pullulan and dextran), animal-free derived materials, and which were extensively studied by our team. Their similar structure with bone extracellular matrix and their high biocompatibility make them promising candidates for GBR procedures. We emphasized those comments according to your suggestion (lines 402-404 and 416-418).
In general, the preparation of the membrane is based on previous studies and some modifications made by the researchers are added.
The experimentation is sound but more studies are necessary to prove that the thin fibrotic capsule has appeared around the implants treated with Mb and Mb + HA, not represent a risk in the future
This study was indeed based on a formulation already described by our team for different types of tissue engineering applications. In the presented work, we applied the formulation that showed the best results in terms of inflammatory reaction and thickness of the fibrotic capsule (75/25 for pullulan/dextran). Moreover, the thin fibrotic capsule disappeared 4 weeks after implantation. This additional comment was made on lines 437-438.
The conclusions are consistent but the study is an early stage.
The study presented here showed preliminary results on the potential of these membranes intended for GBR procedures. Further investigations on a larger animal model (e.g., sinus floor augmentation) is planned.

Reviewer 2 Report
Ensure that all the citations are consistently formatted and that every reference cited in the text is included in your bibliography.
Consider emphasizing the novelty of your work in the broader context of existing literature more explicitly. Other vert similar membranes are already available. Why is this different? Consider emphasizing what sets your membranes apart from existing solutions in terms of biocompatibility.
You should add a Limitations subsection to acknowledge any constraints of your study.
A more detailed discussion on the mechanical properties of the membranes could add value, especially how these properties might influence the GBR process.
Given that the ultimate goal is clinical application, you may want to discuss more explicitly how these findings could be translated into clinical practice.
While you briefly mention that Mb + HA will be further assessed, a more detailed account of planned future work could be beneficial.
Ensure all figures and tables are clearly labeled and referenced in the text.
Your conclusions are clear and directly tied to your research objectives and findings. You might want to underline the broader impact of your work more emphatically here.
Your manuscript is well-structured, making it easy to follow but needs some English revision.
Author Response
Dear Reviewer,
Please consider the responses to the reviewers of our manuscript entitled “Development of novel polysaccharide membranes for Guided Bone Regeneration: in vitro and in vivo evaluations”.
We thank the reviewers for their thorough revision and we consider that the reviewer’s comments have helped to improve substantially the manuscript. We have modified the manuscript accordingly, as explained in detail in the responses to the reviewer and as highlighted in yellow in the main manuscript file (please find attached the corrected manuscript)
We hope that the changes we have made to the manuscript and our answers are in accordance with the reviewer requirements and, as a consequence, that this manuscript could be accepted for publication in “Bioengineering”.
Should you require any additional information and/or clarification, we remain available to provide it to you.
With our best regards,
Naïma AHMED OMAR
-----------------------------------------------------------------------------------------
Comments
Ensure that all the citations are consistently formatted and that every reference cited in the text is included in your bibliography.
We thank the reviewer for this comment. We noticed some inconsistencies in the text and the corrections were made.
Consider emphasizing the novelty of your work in the broader context of existing literature more explicitly. Other vert similar membranes are already available. Why is this different? Consider emphasizing what sets your membranes apart from existing solutions in terms of biocompatibility.
We thank the reviewer for this valuable remark. As said in the introduction, different types of membranes derived from synthetic (e.g., polylactic acid, polylactic-co-glycolic) or natural compounds (e.g., collagen from bovine or porcine origin) are already available in clinics, but mainly collagenic membranes have been used in practice. Due to their poor mechanical properties and their uncontrollable resorption rate, alternatives are required to provide a more suitable environment for bone ingrowth.
Moreover, ethical concerns from the origin of such materials comfort the need for new membranes free from animal origin. To our knowledge, pullulan and dextran were never used in combination to design such GBR membranes. We hypothesized here that thanks to their similar structure to ECM and their good biocompatibility, they could be potential candidates for GBR procedures.
Finally, our team previously described the interest of such blend (shaped into microbeads) as a bone filling material. They highlighted their biocompatibility and their osteoconductive properties in preclinical models (small and large animals). We emphasized those comments according to your suggestion (lines 402-404 and 416-418).
You should add a Limitations subsection to acknowledge any constraints of your study.
The main limitation of this study relies on the femoral defect performed on rats. We discussed the timepoint used here and suggested that later timepoints should be considered in the future for the investigation of more mature bone (lines 478-482).
A more detailed discussion on the mechanical properties of the membranes could add value, especially how these properties might influence the GBR process.
We thank the reviewer for this relevant comment. To improve mechanical properties of a design material, its association with a ceramic (e.g., HA) is often performed. Since blot clot stability during a GBR procedure is a key factor to ensure bone regeneration, a focus on the mechanical strength of the membrane is needed. Additionally, it will maintain bone ingrowth by avoiding membrane collapsing. We then provide a short discussion of the expected mechanical properties for this potential candidate (Mb + HA). (lines 457-464)
Given that the ultimate goal is clinical application, you may want to discuss more explicitly how these findings could be translated into clinical practice.
This study is still at an early stage by investigating bone formation in a small animal model. For its translation to clinical practice, a larger animal model will be required to investigate in a more relevant physiological model its manageability and its potential osteogenic properties. In this purpose, a sinus floor augmentation using a bone graft substitute with the suitable model could be performed to evaluate its efficacy in clinics. A comment was added for this investigation (lines 496-498).
While you briefly mention that Mb + HA will be further assessed, a more detailed account of planned future work could be beneficial.
We thank the reviewer for this suggestion. The membrane reinforced with HA particules appeared to be the best candidate to investigate into more detail its efficacy for GBR procedures. Its implantation in a mandibular defect will permit to assess its potential to boost alveolar bone regeneration (i.e., a more relevant model for its translation into clinics). A future plan on the ongoing experiment is provided to assess the use of Mb + HA for GBR procedures in the conclusion part (lines 495-497 and 513-514).
Ensure all figures and tables are clearly labeled and referenced in the text.
We corrected the mistakes in labeling the figures and we introduced them accordingly in the text.
Your conclusions are clear and directly tied to your research objectives and findings. You might want to underline the broader impact of your work more emphatically here.
We thank the reviewer for this comment. In the conclusion part, we outlined our future research plan to further validate potential future assessments of Mb + HA (lines 513-514).
Your manuscript is well-structured, making it easy to follow but needs some English revision.
We thank the reviewer for this suggestion. In the whole revised text, we improve the English quality. Corrections for language quality were not highlighted for better comprehension in the text.
